# Acanthopanax Senticosus Saponins Prevent Cognitive Decline in Rats with Alzheimer’s Disease

**DOI:** 10.3390/ijms26083715

**Published:** 2025-04-14

**Authors:** Xue-Min Cui, Wang Wang, Lin Yang, Bao-Wen Nie, Qian Liu, Xiao-Hui Li, Dong-Xiao Duan

**Affiliations:** Department of Physiology and Neurobiology, School of Basic Medical Sciences, Zhengzhou University, Zhengzhou 450001, China; 18331066539@163.com (X.-M.C.); zzuwwuzz@163.com (W.W.); yanglinlalala233@163.com (L.Y.); niebaowen@gs.zzu.edu.cn (B.-W.N.); 15256520363@163.com (Q.L.); lxhui122@163.com (X.-H.L.)

**Keywords:** Tau protein, Alzheimer’s disease, cognitive decline, Acanthopanax senticosus saponins, inflammation

## Abstract

Alzheimer’s disease (AD) is a progressive degenerative disease of the nervous system that affects older adults. Its main clinical manifestations include memory loss, cognitive dysfunction, abnormal behaviour, and social dysfunction. Neuroinflammation is typical in most neurodegenerative diseases, such as AD. Therefore, suppressing inflammation may improve AD symptoms. This study investigated the neuroprotective effects of *Acanthopanax senticosus saponins (ASS)* in an AD model induced by streptozotocin (STZ). Here, we characterised a rat model of STZ-induced AD with the parallel deterioration of memory loss and neuroinflammation. Following the end of the treatment with ASS (50 mg/kg for 14 consecutive days), behavioural tests (Morris water maze test, Y-maze test) were performed on the rat, and the molecular parameters (DAPK1, Tau5, p-Tau, NF-κB, IL-1β, TNF-α, and NLRP3) of the rat hippocampus were also assessed. We demonstrated that ASS, which has potent anti-inflammatory effects, can reduce neuroinflammation and prevent cognitive impairment. In the water maze test, ASS-treated groups exhibited significantly increased average escape latency (*p* < 0.05), the percentage of stay in the target quadrant (*p* < 0.05), and the number of times each group of rats crossed the platform (*p* < 0.05) compared to the negative control. And ASS could reduce the phosphorylation of the Tau protein (*p* < 0.001) and death-associated protein kinase 1 (DAPK1, *p* < 0.001) in the hippocampal tissue, improving cognitive impairment in STZ-treated rats by suppressing the inflammatory response; the molecular analysis showed a significant reduction in pro-inflammatory markers like NLRP3, IL-1β, TNF-α, and NF-κB (*p* < 0.001). It was also discovered that the NF-κB inhibitor SN50 had the same effect. Therefore, the present study used ASS through its anti-inflammatory effects to prevent and treat AD. This study highlights the potential efficacy of ASS in alleviating cognitive dysfunction in AD.

## 1. Introduction

Alzheimer’s disease (AD) is a typical degenerative disease that mainly involves learning and memory loss associated with cognitive impairment, such as reduced spatial discrimination ability, often characterised by progressive memory, learning, and cognitive dysfunction [1,2]. The main pathological features of AD are extracellular beta-amyloid protein aggregation and excessive deposition, forming starch spots [3]. Intracellular hyperphosphorylation of the Tau protein leads to neurofibrillary tangles, which seriously affects the normal life of patients [4,5]. Regrettably, the mechanism is still unclear to date. Currently, there is no cure for these diseases, and the existing therapeutic approaches mainly focus on symptom management. However, these treatments often have limited efficacy and may be associated with significant side effects. Therefore, there is an urgent need to develop novel therapeutic strategies to halt or reverse the progression of neurodegeneration. In recent years, more and more studies have found a certain correlation between chronic inflammation and cognitive dysfunction. Inflammation can cause neuronal damage and neurofibrillary tangles, affecting the normal function of the nervous system and leading to cognitive dysfunction. Additionally, inflammatory cytokines play a key role in the occurrence and development of cognitive dysfunction. Inflammatory cytokines include tumour necrosis factor, interleukin, chemokines, etc., which can enter the central nervous system through the blood–brain barrier and have toxic effects on neuronal cells, resulting in decreased neuronal function and death, thereby reducing cognitive function. Moreover, there is a bidirectional regulation between inflammation and cognitive dysfunction. Cognitive dysfunction can also aggravate the inflammatory response. On the one hand, cognitive dysfunction may lead to a decrease in patients’ quality of life and increase the risk of infection, thereby triggering or worsening inflammation. On the other hand, cognitive dysfunction may affect patients’ immune function, resulting in a weakened response to inflammation by the body, making it difficult to control inflammation. Oxidative stress plays an important role not only in type 2 diabetes but also in AD and related nervous system diseases. In cells, oxidative stress is caused by the disruption of the balance between the production of reactive oxygen species/reactive nitrogen species and the cellular defence provided by antioxidants. Inflammation is a prominent feature of many chronic diseases, including AD and DM [6].

The current AD treatments only aim to improve clinical symptoms or postpone cognitive decline. The main drugs currently approved by the Food and Drug Administration for treating AD are acetylcholinesterase inhibitors (donepezil, galantamine, and carbalatin) and N-methyl-D-aspartate receptor blockers (memantine). These drugs can only alleviate symptoms and do not delay or stop AD progression; these synthetic drugs also have certain side effects [7,8,9]. Furthermore, AD treatment has focused on Aβ for many years with no successful results. Therefore, researchers believe that Tau pathology cannot be ignored in AD aetiology, and decreased Tau protein levels have been suggested as a potential therapeutic strategy for AD [10,11].

Chinese herbal medicine has multi-target, multi-link, and multi-path comprehensive regulation and treatment effects and has received increasing attention in AD treatment [8,12]. The use of medicinal plants has developed in various periods of human history. The special interest in the antioxidant, anti-inflammatory, and antiviral potential of medicinal plant extracts is sparked by the presence of specific secondary metabolites [13]. However, the exact cause of Alzheimer’s disease has remained unclear since its discovery. The use of traditional Chinese medicine to treat AD has several advantages [12,14]. *Acanthopanax senticosus* is a plant of the Acanthopanax family, with effects similar to those of “Fuzheng Guiben, Yizhi Anshen, and Bushen Jianpi”. The *Compendium of Materia Medica* regards them as “high-grade goods”. Acanthopanax senticosus saponins (ASS), which include triterpenoid saponins, such as saponins A, B, C, and E, are metabolised through hepatic oxidative conjugation, enterohepatic recirculation, and gut-microbiota-mediated hydrolysis. These processes ultimately convert the parent compounds into aglycones, oxidised triterpenoid derivatives, and glucuronide conjugates. The resulting metabolites exert dual antioxidant and anti-inflammatory effects by suppressing NF-κB-mediated inflammatory signalling and activating Nrf2-dependent antioxidant pathways [15,16]. Pharmacological studies have shown that ASS exerts immunomodulatory, anti-stress, anti-inflammatory, anti-fatigue, antioxidant, and neuroprotective effects [16,17,18]. However, recent studies have shown that *Acanthopanax senticosus* regulates cardiovascular and cerebrovascular systems, prevents brain ageing, and promotes learning and memory functions. In addition, some studies have shown that ASS can inhibit chronic inflammation and ageing and may preserve cognitive function [16,19,20]. However, its effects on Tau phosphorylation in patients with AD are yet to be reported.

Chain urea with cephalosporins streptozotocin (STZ) is a Nokia nitrourea product that can impair the insulin receptor signal transduction pathways, reduce glucose use in rats due to cerebral energy metabolism disorder, and decrease ATP and GTP generation caused by the cortex-associated cognitive or neural function damage, affecting synaptic transmission and plasticity and leading to learning and memory dysfunction [21,22,23]. This study established a rat model of AD using STZ, and ASS was intraperitoneally injected for intervention [24]. The Morris water and Y-mazes were used to detect changes in learning and memory and to explore the effects of ASS on the learning and memory function of the rat model of AD induced by STZ. We aimed to test whether ASS had neuroprotective and restorative properties using an STZ-induced rat model of AD. Simultaneously, we explored whether factors associated with altered cognitive impairment were associated with Tau protein phosphorylation and inflammatory factors.

## 2. Results

### 2.1. AD Model Rats Induced by STZ Show Cognitive Dysfunction

The animal model was developed as previously described. Assessment of cognitive function in different groups of rats was performed using the water maze and Y-maze tests. In the water maze test, escape latency during the first 5 days [F(1,80) = 11.08], the number of times the rats crossed the target quadrant [F(9,9) = 2.868], and the percentage of time in the target quadrant [F(9,9) = 2.088] were assessed. The MWMT revealed significant differences in total swimming distance and escape latency between the two groups during the first 5 days (Figure 1A, *p* < 0.05). On the sixth day, the number of platform crossings and the percentage of time in the target quadrant were significantly lower in the depression group (Figure 1B–D, *p* < 0.01, *p* < 0.01). These results showed that spatial exploration and navigational ability decreased in STZ-treated rats, indicating impaired spatial memory. The alternate correct rat rate (F(1,9) = 33.94) was evaluated for each arm in the Y-maze test. Rats in the STZ-treated group had a reduced alternate correct rate in the Y-maze test compared with normal controls (Figure 1E, *p* < 0.01), indicating reduced spatial working memory capacity and cognitive dysfunction.

### 2.2. ASS Can Improve Cognitive Dysfunction in STZ-Treated Rats

A total of 30 rats were collected and 20 were prepared as an STZ-treated model. The STZ model rats were divided into two groups: the STZ-treated and STZ + ASS groups. The STZ + ASS group was intraperitoneally injected with ASS for 2 weeks. Similarly, the control and STZ-treated rats were injected with the same amount of normal saline for 14 days, and the behavioural test results were compared. In the water maze test, escape latency during the first 5 days [F(2,120) = 10.80], the number of times the rats crossed the target quadrant [F(2,27) = 11.78], and the percentage of time in the target quadrant [F(2,27) = 9.915] were assessed. The MWMT showed that the total swimming distance and escape latency of rats in the control, STZ-treated, and STZ +ASS groups were significantly different during the first 5 days (Figure 2A, *p* < 0.05). On the sixth day, the number of platform crossings (*p* < 0.05) and time spent in the target quadrant (*p* < 0.05) were significantly higher in the STZ + ASS group (Figure 2B–D). These findings indicated that the STZ +ASS group exhibited significant improvements in terms of spatial exploration and navigation. The alternate correct rat rate (F(2,18) = 1.209) was evaluated for each arm in the Y-maze test. The working memory of the rats in each group was examined using the Y-maze experiment (Figure 2E). The results showed no significant differences among the three groups, which may be due to the large individual differences in the mice in this group, the underlying causes of the varying degrees of inflammatory response observed at different time points, and the same group of animals being tested over a longer period of time. Further research is required to gain a deeper understanding of this topic.

### 2.3. ASS Causes Neuronal Changes in the Hippocampal Region

We stained neurons in the CA3 region of the hippocampus using immunofluorescence and HE techniques. Cone cells in STZ-treated rats were sparse compared with the control group, where cells were neatly and tightly arranged, and the number of cells was reduced in the CA3 regions, which emerged especially prominently in immunofluorescence experiments. The number of neurons in the corresponding regions of the ASS group was higher than that in STZ-treated rats (Figure 3). This indicates a reduced number of neurons in STZ-treated rats due to the cognitive dysfunction effect of the drug, which may be associated with a damaged or reduced number of neurons. ASS may inhibit neuroinflammation and reduce the release of inflammatory factors, protecting nerve cells and alleviating AD.

### 2.4. Effects of ASS on the Hyperphosphorylation of Tau Protein in the Hippocampus of STZ-Treated Rats

In AD, cognitive impairment and the hyperphosphorylation of Tau proteins are inextricably linked, confirming that ASS improves cognitive impairment in association with Tau protein expression. In the Tau protein expression assay conducted in the hippocampus, the molecular estimates of Tau5, pT231, pSer262, and pSer396 exhibited statistically significant differences (F(2,12) = 98.15, F(2,12) = 31.12, F(2,12) = 74.72, and F(2,12) = 62.14, respectively). In the STZ-treated rats, Tau5 expression decreased (*p* < 0.001) in the murine hippocampus region after ASS treatment (Figure 4A,B). Simultaneously, Tau phosphorylation at Thr231, Ser262, and Ser396 decreased (*p* < 0.001) (Figure 4A,C–E). This indicates that reduced Tau5 expression and phosphorylation improve cognitive dysfunction in STZ-treated rats. This suggests that ASS alleviates cognitive dysfunction in STZ-treated rats by reducing Tau protein phosphorylation.

### 2.5. Effects of ASS on the STZ-Induced Inflammatory Cytokines in STZ-Treated Rats

Death-associated protein kinase 1 (DAPK1) is a protein kinase that phosphorylates serine and threonine residues at protein sites, and our group demonstrated that DAPK1 phosphorylates the Tau protein, leading to cognitive decline in the mouse models of AD. Therefore, the expression of DAPK1 and the related protein in the hippocampal region was detected using Western blotting. The molecular estimates of DAPK1, NLRP3, NF-κB, lL-1β, and TNF-α were statistically significantly different (F(2,12) = 36.23, F(2,12) = 23.13, F(2,12) = 70.81, F(2,12) = 18.97, and F(2,12) = 15.77). And it was found that the protein expression of rats in the ASS group was significantly decreased (*p* < 0.001) compared with that in the STZ-treated rats (Figure 5A,B). However, the expression of some inflammation-related markers—NLRP3 (*p* < 0.001), NF-κB (*p* < 0.001), lL-1β (*p* < 0.001), and TNF-α (*p* < 0.001)—appeared consistent with the expression of DAPK1 (Figure 5A,C–F), suggesting that elevation of DAPK1 might be associated with the altered expression of inflammatory factors.

### 2.6. DAPK1 Protein Expression in the Hippocampal Region of STZ-Treated Rats Is Associated with NF-κB

To investigate whether the effects of ASS are related to anti-inflammation, we used SN50—an inhibitor of NF-KB—to demonstrate the relationship between inflammatory factors and DAPK1. The molecular estimates for NF-κB, DAPK1, and pS396 Tau proteins were statistically significantly different (F(2,6) = 83.32, F(2,6) = 54.78, and F(2,6) = 156.4). And we found a reduced expression of DAPK1 (*p* < 0.001) (Figure 6A,C). Tau protein phosphorylation at Ser396 showed the same changes (*p* < 0.001) (Figure 5A,D). This suggests that the elevated DAPK1 expression in this model may be associated with the expression of inflammatory factors.

## 3. Discussion

Neurodegeneration is a complex process involving multiple molecular mechanisms. Oxidative stress, mitochondrial dysfunction, and neuroinflammation are among the known mechanisms contributing to neurodegeneration. For example, neuroinflammation triggers an immune response, causing neuronal damage. These mechanisms interact and contribute to the progressive loss of neurons in neurodegenerative diseases. Anti-inflammatory agents are employed to reduce neuroinflammation. ASS has recently attracted increasing attention because it reduces the level of lipid peroxide in the rat brain, increases the activity of superoxide dismutase in the plasma, enhances the scavenging ability of free radicals, strengthens cell membrane stability, protects neurons, and improves synaptic function [15,17]. However, there are few reports on the protective effects of ASS in the central nervous system. This study found that ASS could improve STZ-induced cognitive deficits in rats by regulating DAPK1 expression in hippocampal tissues and inhibiting the over-phosphorylation of the Tau protein while simultaneously reducing the expression of neuroinflammatory factors in hippocampal tissues. The trend of the changes between the phosphorylation of Tau protein, elevated expression of DAPK1 protein, and increased inflammatory factors was consistent, and this change should be further explored, as the results may provide new ideas and approaches for ASS as a potential candidate for AD treatment.

STZ was first used to prepare diabetes models, a commonly used and well-established animal model [21,22]. However, it has recently been found that injecting STZ into the brain’s lateral ventricle can cause cognitive dysfunction in rats. The test results confirm that the model displays AD symptoms due to the hyperphosphorylation of the Tau protein [24,25]; therefore, this model is gradually becoming useful in the study of cognitive dysfunction. The study of this model also found an increased expression of inflammatory factors [26,27]. However, it remains unclear whether this change was associated with cognitive dysfunction due to the increased phosphorylation of the Tau protein and its mechanism. The present study found that rats in the STZ-treated group had decreased spatial orientation and exploration memory and showed obvious cognitive dysfunction, which could be alleviated and reversed by ASS, as shown using the water maze and Y-maze tests. Cognitive dysfunction was associated with hyperphosphorylation of the Tau proteins at the Thr231, Ser262, and Ser396 sites. Previous research by our group found that DAPK1 can phosphorylate these sites and is crucial in cognitive dysfunction in patients with AD or AD mouse models [28,29]; therefore, the protein expression of DAPK1 was significantly elevated in STZ-treated rats. This suggests that phosphorylation of the Tau protein is achieved through elevation of DAPK1. However, the effect of ASS, an anti-inflammatory drug, on the expression of DAPK1 and Tau and the relationship between the three need to be further explored.

*Acanthopanax senticosus (AS)* is a relatively well-defined and anti-inflammatory herbal component, and studies have reported that AS has various pharmacological activities, such as scavenging free radicals to increase superoxide dismutase activity in the brain, inhibiting endothelin secretion and nitric oxide release, decreasing neuronal apoptosis during ischaemia, and increasing neuronal survival [30]. Jung et al. found that the pharmaceutical components extracted from AS have anti-inflammatory and analgesic effects [31]. Similarly, Su et al. found that the extract of AS effectively treats liver injury and enhances the body’s antioxidant capacity [32]. Feng et al. explored the mechanism of action of ASS on AD using network pharmacology and molecular docking techniques. The molecular docking results showed that ASS B and D1 have a strong binding ability, suggesting that spikenard is feasible in AD treatment [33]. Lu et al. found that a combination of baixantin, which contains ASS, protects the substantia nigra and striatum from oxidative stress and inflammatory factors, ultimately acting as an anti-Parkinson’s disease agent [34]. A previous study reported that an AD model prepared using STZ showed elevated brain levels of inflammatory factors. Notably, some inflammatory mediators have been used as AD markers, such as TNF-α, IL-1β, Iba-1, GFAP, NF-κB, TLR2, and MHCII, and the use of anti-inflammatory drugs has become a potential therapeutic approach for AD [35,36,37].

Furthermore, it has been elucidated that ASS has an anti-neuroinflammatory effect; however, its mechanism of action still needs to be explored. Therefore, we used SN50, an inhibitor of NF-κB, to act on the hippocampal CA3 region with increased DAPK1 expression in STZ-treated rats and found that DAPK1 expression was reduced by inhibiting NF-κB expression. However, phosphorylation of the Ser396 site of the Tau protein was similarly decreased, suggesting that the increase in inflammatory factors indirectly affects the decline in cognitive function in STZ-treated rats, and it can be influenced by altering the DAPK1 expression, leading to decreased Tau protein phosphorylation and alleviated cognitive dysfunction in STZ-treated rats.

Therefore, in this study, ASS reduced the expression of inflammatory factors through its anti-inflammatory effects and inhibited the expression of DAPK1, and there was a tendency for the number of hippocampal neurons to increase. DAPK1 has an important role in apoptosis, and it is not known whether the improvement in the number of neurons in the ASS group was caused by a weakening of the destructive effect of DAPK1 or an enhancement of the protective effect of ASS, and further experiments are needed to prove this. It is clear that reduced expression of DAPK1 significantly reduces the phosphorylation of Thr231, Ser396, and Ser262 sites in the rat hippocampus and ultimately improves cognitive dysfunction in STZ-treated rats. These findings suggest that the inhibitory effect of ASS on these specific hyperphosphorylation sites of Tau proteins is one of the mechanisms underlying its ameliorating effect on cognitive impairment. The pathway through which inflammatory factors regulate DAPK1 and the specific mechanism should be explored in future experiments.

## 4. Materials and Methods

### 4.1. Animals

Male SD rats (8 weeks old, 220 ± 15 g) were obtained from the Henan Experimental Animal Center (license: SCXK 2021-0011) and housed in a temperature-controlled room (22 ± 0.5 °C). Notably, all rats were maintained on a 12 L: 12 D schedule, with lights on at 7 a.m. and off at 7 p.m., with ample food and water. Behavioural testing began at 8:00 a.m. This experiment was conducted by the same worker from animal rearing to behavioural testing, and the light, environment, and working conditions in the room were uniform during the experiment; therefore, there was no interference with the behavioural measurements of the rats. The experimental procedures followed the Ethical Principles in Animal Research adopted by the Ethics Committee on the Use of Animals of Zhengzhou University (Ethic NO. ZZULAC2022-118). All experiments followed good laboratory practice protocols and quality assurance methods.

### 4.2. Model Development and Treatment

Male SD rats (n = 30) were divided into three groups, namely the control, STZ-treated, and STZ + ASS groups. After weighing, the rats were anaesthetised with 2% isoflurane and 0.3% pentobarbital sodium (1 mL/100 g) and fixed using a brain stereolocator, routine skin disinfection, midsagittal incision, and a flexible skull drill. The posterior–anterior skull was 0.8 mm; the right side of the sagittal suture was opened by 1.5 mm; and the brain surface was lowered vertically by 3.6 mm. The control group were injected with cerebrospinal fluid; the STZ-treated and STZ + ASS groups were injected with STZ (120 mg/mL, 3 μL/100 g; Cat. No. S0130, Sigma, St. Louis, MO, USA); and the skin was sutured using a needle and line. Gentamicin sulphate was applied to the wound for 3 consecutive days to prevent infection. After 2 weeks of feeding, rats were randomly selected from the control and STZ-treated groups (model control and treatment groups, respectively), and rats were subjected to the Morris water and Y-maze tests. Rats in the model treatment group were intraperitoneally injected with 5% ASS (Ciwujia extract complex, LMAI Bio, Shanghai, China) at 50 mg/kg for 14 consecutive days. The control and STZ-treated rats were continuously injected with the same amount of normal saline for 14 days.

The same method was used to prepare the STZ model, which was divided into two groups. One group were injected with cerebrospinal fluid in the hippocampal CA3 region. The other group were injected with SN50, an inhibitor of nuclear factor kappa (NF-κB), defining fontanel as the origin of the coordinates (0 mm, 0 mm, 0 mm), resulting in an injection point of (±3 mm ML, −2.5 mm AP, −2.5 mm DV). The ultimate concentration of SN50 was 3.595 μM, and each side of the hippocampus was injected with 0.8 μL of the configured agent at a rate of 0.01 μL/min. The next experiment was conducted after 2 weeks of observation.

### 4.3. Behavioural Tests

#### 4.3.1. Morris Water Maze Test (MWMT)

Rats were transferred to the behavioural test room in advance for habituation, lasting approximately 20 min. The Morris water maze test is divided into two periods that are used to detect changes in spatial learning and spatial memory [28]. Behavioural tests were performed after 2 weeks of administration. The tests were classified as follows: (1) The navigation experiment: A third-place platform was placed in the fourth quadrant from the wall, and then, each rat was released into the water from the four marked points facing the wall. Observation records were performed to indicate whether the rats could find and climb up to the platform roadmap within 90 s. This is the time taken to escape the incubation period. If the rats did not find the platform within 90 s, the experiment was automatically stopped, and the escape latency was set at 90 s. The rats were drawn to the platform, kept for 30 s, and returned to the cage. If the rats found the platform, the cameras were immediately stopped, and the escape incubation time was recorded. The rats were allowed to remain on the platform for 30 s. The training lasted 5 days. (2) The space exploration experiment: The platform was removed on the 6th day after the positioning navigation experiment, and the rats were placed in water at the entry point of the second quadrant farthest from the platform. The retention time of the rats in the quadrant where the platform was placed was observed for 90 s, and the number of times the rats crossed the platform was recorded as the memory score.

#### 4.3.2. Y-Maze

The Y-maze test was used to detect changes in learning and memory [28].

The three arms of the Y-maze were labelled as 1, 2, and 3. During the test, ambient noise was minimised. The animals were placed in the maze for 2 h before the formal experiment, with a 5 min test time for each mouse. Each mouse was then placed at the intersection of the maze’s three arms, ensuring that their heads were in the same direction. The mice’s total number and arm orders were recorded using Smart 3.0.05. The spontaneous alternation accuracy of each mouse was calculated based on the total number and sequence of arm entries using this formula: (total correct arm entry sequence/total arm entry times − 2) ÷ 100%, where mice continuously entering three different arms are regarded as one accurate alternate.

### 4.4. Sample Preparation and Molecular Biology Experiments

After behavioural experiments were performed, the rats were deeply anaesthetised with 4–5% isoflurane (RWD, Shenzhen, China). Then, some of the rats were decapitated to obtain fresh hippocampal tissues for Western blotting (each mouse’s left and right hippocampus were put together as a group), and cardiac perfusion was conducted for the other rats with 0.9% saline and 4% paraformaldehyde (PFA) sequentially to obtain fixed brain tissues, which were used for HE staining and immunofluorescent staining.

#### 4.4.1. Haematoxylin and Eosin (HE) Staining

Frozen sections (15 μm) from three randomly selected rats in each group were stained in haematoxylin solution for 2 min. The slices were then decolourised in acid alcohol (1% hydrochloric acid (HCL)) for 15 s in an HCL–ethanol differentiation liquid, washed in running tap water, and stained with eosin for 1 min. The samples were examined under a microscope after sealing with a neutral resin.

#### 4.4.2. Immunofluorescent Staining

After blocking with 5% bovine serum albumin (Proteintech, Wuhan, China) containing 1% Tririon X-100 for 1 h at 23 °C, hippocampal sections were incubated at 4 °C with the primary antibody overnight (NeuN, rabbit, 1:200, service-bio, Wuhan, China, Cat# GB11138). Then, the slices were washed and incubated with secondary antibodies—Alexa Fluor^®^ 488 goat anti-mouse IgG H&L (1:1000, Abcam, Cambridge, UK, Cat# ab150113, RRID: AB_2576208)—for 2 h at 23 °C. The sections were then mounted on slides and sealed with a neutral resin. Images were acquired using a laser confocal microscope (FV1000 OLYMPUS, Tokyo, Japan).

#### 4.4.3. Western Blot

For the ASS treatment experiments, the sample size was set to 5, while for the SN50 treatment experiments, it was set to 3. One hippocampus from each animal was used individually in the same group. The rat hippocampal tissue was crushed using an ultrasonic cell disintegrator after adding phenylmethanesulfonyl fluoride (PMSF) to inhibit protease activity and radioimmunoprecipitation assay (RIPA) lysate buffer. The rat hippocampus was centrifuged at 12,000 rpm for 10 min; the supernatant was extracted; and the protein concentration was determined using a BCA protein assay kit [28]. The protein mixture was boiled for 10 min in a 100 °C water bath for denaturation before the sample was collected. An amount of 20-40 μg protein was loaded in each well of a 10% sodium dodecyl sulphate-polyacrylamide separation gel (9 wells, 1 mm thickness), and gel electrophoresis (100 V, 150 min) was performed. The proteins were transferred to a polyvinylidene fluoride (PVDF) membrane (200 mA, 60 min) after electrophoresis. The membranes were then immersed in 5% skim milk powder and blocked for 2 h. After closure, the primary antibody was added, including Phospho-Tau [Ser 396] (rabbit, 1:1000, Abcam, Cambridge, UK, Cat# ab109390, RRID: AB_10860822), Phospho-Tau [Ser 262] (rabbit, 1:500, Abcam, Cambridge, UK, Cat# ab131354, RRID: AB_11156689), Phospho-Tau [Thr 231] (rabbit, 1:1000, Abcam, Cambridge, UK, Cat# ab151559, RRID: AB_2893278), total Tau (mouse, 1:1000, Abcam, Cambridge, UK, Cat# ab80579, RRID: AB_1603723), DAPK1 (rabbit, 1:1000, Abcam, Cambridge, UK, Cat# ab193442), NLRP3 (rabbit, 1:1000 ABclonal, Wu-Han, China, Cat# A12694, RRID: AB_2759538), NF-κB (rabbit, 1:1000, Abcam, Cambridge, UK, Cat# ab16502, RRID: AB_443394), lL-1β (rabbit, 1:1000, ABclonal, Wu-Han, China, Cat# A1112, RRID: AB_2758416), TNF-α (rabbit, 1:1000, service-bio, Wu-Han, China, Cat# GB112188), and β-actin (mouse, 1:10,000, Proteintech, Wu-Han, China, Cat# 66009-1-Ig), and incubated overnight at 4 °C. The next day, goat anti-rabbit or goat anti-mouse IgG H&L (HRP) (1:2000, CST, Boston, USA, Cat# 7076S) or secondary antibodies—goat anti-rabbit (1:2000, CST, Boston, USA, Cat# 7074S)—were incubated at room temperature for 2 h. Furthermore, the chemiluminescent liquid was prepared with the ECL Western Blot kit, and exposure imaging was performed using a chemiluminescent imager. The protein intensity was calculated using ImageJ software (imagej.orgJava 1.0.8-345 (64-bit)). The intensity of the bands was quantified, and the values were normalised to the beta-actin levels.

### 4.5. Statistical Analysis

Data were analysed using GraphPad Prism 8.0.2 (GraphPad Software, La Jolla, CA, USA, www.graphpad.com) and are expressed as mean ± standard error (SE). Two-way analysis of variance with repeated measures followed by Tukey’s post hoc test was conducted to assess escape latency and total distance and latency to the target in the Morris water maze test during the training period. One-way analysis of variance followed by Tukey’s post hoc test or unpaired *t*-test was performed for other group comparisons as appropriate. A *p*-value < 0.05 was considered statistically significant. Statistical analysis was conducted by an evaluator blinded to the group assignments.

## 5. Conclusions

STZ can produce a simple and reliable AD model and detect hyperphosphorylation of the Tau protein in the hippocampus, leading to a decline in cognitive function in model rats. After treatment with ASS, the expression of inflammatory cytokines and DAPK1 in the hippocampus was reduced, and the learning and memory functions of STZ-treated rats improved, suggesting that ASS can reduce the phosphorylation of the Tau protein and improve memory function by inhibiting the expression of DAPK1, which is elevated in NF-κB. These findings provide a scientific basis for the potential application of ASS in the clinical treatment of AD.

## Figures and Tables

**Figure 1 ijms-26-03715-f001:**
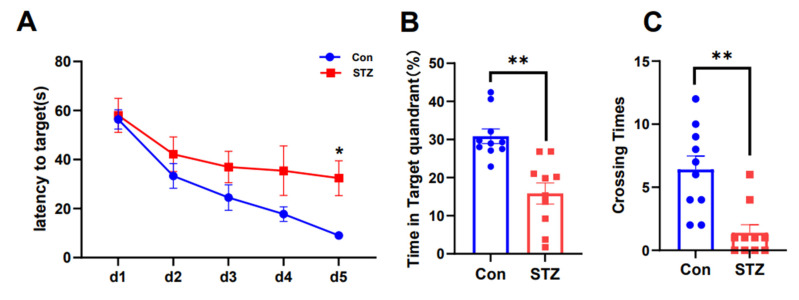
Cognitive dysfunction in STZ-treated model rats. Morris water maze test and Y-maze test (n = 10/group). (**A**) Escape latency during the first 5 days; (**B**) Number of times the rats crossed the target quadrant; (**C**) Percentage of time in the target quadrant in both groups; (**D**) Representative swim paths in the test session on day 6. * *p* < 0.05, ** *p* < 0.01 compared with control); (**E**) Y-maze experiment alternating the correct rate of rats (** *p* < 0.01 compared with the control).

**Figure 2 ijms-26-03715-f002:**
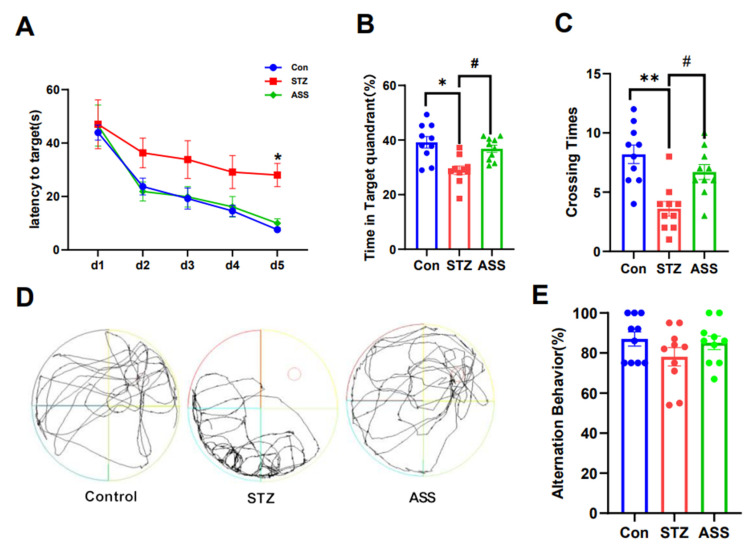
Effect of ASS on the cognitive ability of STZ-treated rats. Morris water maze test and Y-maze test (n = 10/group). (**A**) Average escape latency in the first 5 days; (**B**) The number of times each group of rats crossed the platform; (**C**) The percentage of time in the target quadrant for each group of rats; (**D**) Representative trajectory maps of spatial exploration for each group of rats. * *p* < 0.05, ** *p* < 0.01 compared with the control group. # *p* < 0.05 compared with STZ-treated rats; (**E**) Y-maze experiment alternating the correct rate of rats (* *p* < 0.05 compared with the control group).

**Figure 3 ijms-26-03715-f003:**
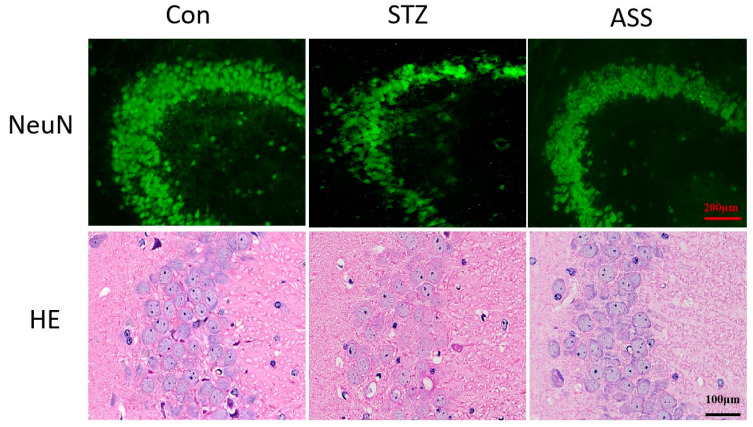
IF and HE staining of rat hippocampus in the CA3 areas. The number of neurons in the CA3 areas was reduced in the STZ-induced rat model of AD, and the number of neurons returned to normal after ASS treatment (n = 3).

**Figure 4 ijms-26-03715-f004:**
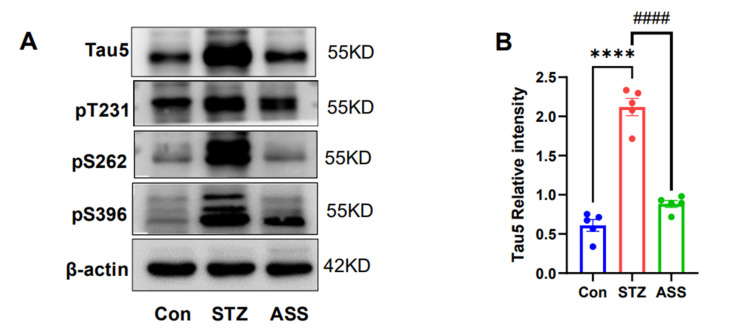
Effect of ASS on the expression of pT231, pSer262, pSer396, and Tau5 in the hippocampal region of STZ-treated rats. (**A**–**E**) Expression of Tau5, pT231, pSer262, and pSer396 proteins in the hippocampal region (n = 5). **** *p* < 0.001 compared with the control group. ^####^ *p* < 0.001 compared with the STZ + ASS group.

**Figure 5 ijms-26-03715-f005:**
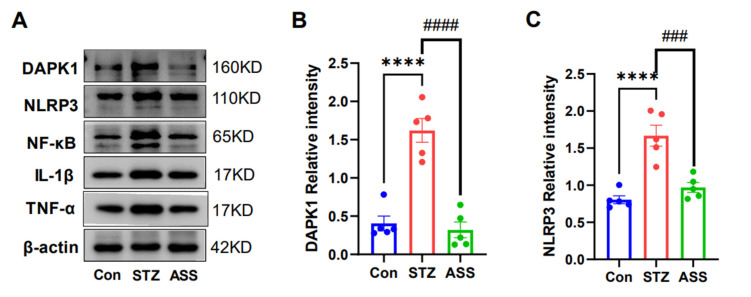
Effect of ASS on the expression of DAPK1, NLRP3, NF-κB, lL-1β, and TNF-α protein in the hippocampal region of STZ-treated rats. (**A**–**F**) Expression of DAPK1, NLRP3, NF-κB, lL-1β, and TNF-α protein in the hippocampal region (n = 5)., *** *p* < 0.001, **** *p* < 0.0001 compared with the control group., ^##^ *p* < 0.01, ^###^ *p* < 0.001, ^####^ *p* < 0.0001 compared with the STZ + ASS group.

**Figure 6 ijms-26-03715-f006:**
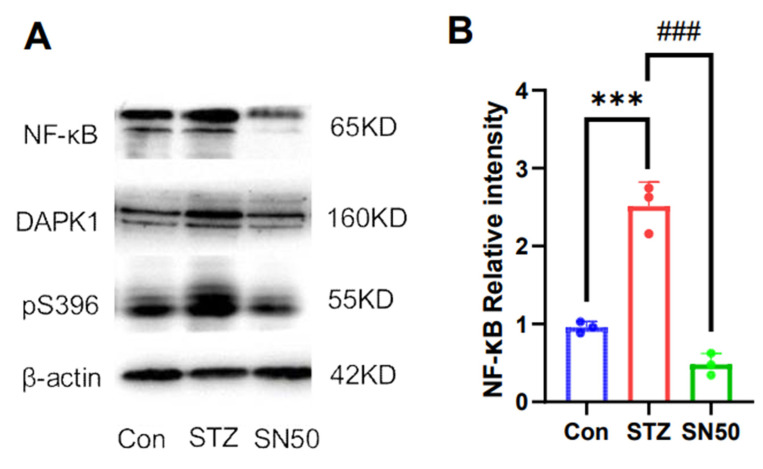
Effects of NF-κB inhibitors on DAPK1 and phosphorylated Tau proteins. (**A**–**D**) Expression of NF-κB, DAPK1, and pS396 Tau proteins in the hippocampal region (n = 3). *** *p* < 0.001 compared with the control group. ### *p* < 0.001 compared with the SN50 group.

## Data Availability

Data is contained within the article.

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
