# Peer review of "Acanthopanax Senticosus Saponins Prevent Cognitive Decline in Rats with Alzheimer’s Disease"

_ijms, 2025, doi:10.3390/ijms26083715_

Round 1
Reviewer 1 Report
Comments and Suggestions for Authors
Dear Authors,
The MS “Acanthopanax senticosus saponins prevents cognitive decline in rats with Alzheimer’s disease” is original research topic which can be accepted for the publishing after minor revision. Please, see some comments bellow
Before line 70 of the Introduction section, it would be beneficial to include information not only about traditional Chinese medicine but also about the ongoing development of methods for identifying specific secondary metabolites and understanding their role in immunomodulatory plant responses and adaptive reactions to various stresses. Please refer to the following sources:
Sytar, O. and Hajihashemi, S. (2024). Specific Secondary Metabolites of Medicinal Plants and Their Role in Stress Adaptation. In Plant Secondary Metabolites and Abiotic Stress (eds G.C. Nikalje, M. Shahnawaz, J. Parihar, H.A. Qazi, V.N. Patil and D. Zhu). https://doi.org/10.1002/9781394186457.ch15
- Yuca, B. Aydin, S. Karakaya, G. Goger, Z. Bingöl, A. Civas, M. Koca, B. Demirci, O. Sytar, I. Gulcin, Z. Guvenalp, Chem. Biodiversity2024, 21, e202301753. https://doi.org/10.1002/cbdv.202301753
I think the latin name of Acanthopanax senticosus must be done in cursive in the text.
L91-92 it would be better to write as aim but not like statement.I would recommend to rewrite this sentence: “This study established a rat model of AD using STZ, and ASS was intraperitoneally injected for intervention.”
In the Material nad methods it is needed to write information about how was received and from where Acanthopanax senticosus saponins (ASS)and from where was received STZ. It was donme special extraction of Acanthopanax senticosus saponins?
L274-275 “ASS can inhibit neuroinflammation and reduce the release of inflammatory factors, protecting nerve cells and alleviating AD.”It would be good to add reference for it or try to use not “can” but “may”
L369-372 please, add reference for this sentence “studies have reported that AS has various pharmacological activities, such as scavenging free radicals to increase superoxide dismutase activity in the 370 brain, inhibiting endothelin secretion and nitric oxide release, decreasing neuronal apoptosis during ischaemia, and increasing neuronal survival.”
Author Response
Point 1: Before line 70 of the Introduction section, it would be beneficial to include information not only about traditional Chinese medicine but also about the ongoing development of methods for identifying specific secondary metabolites and understanding their role in immunomodulatory plant responses and adaptive reactions to various stresses. Please refer to the following sources:
Author response: Thanks for your beneficial suggestion. We have added the requested information about the ongoing development of methods for identifying specific secondary metabolites and their role in immunomodulatory plant responses and adaptive reactions to various stresses line 62 and line 86 of the Introduction section. We have also referred to the provided sources.
Point 2: The latin name of Acanthopanax senticosus must be done in cursive in the text.
Author response: We have ensured that the latin name of Acanthopanax senticosus is presented in cursive in the full text.
Point 3: In lines 91-92, the sentence should be written more as an aim rather than a statement. The reviewer recommends rewriting the sentence: “This study established a rat model of AD using STZ, and ASS was intraperitoneally injected for intervention.”
Author response: As you mentioned, we have rewritten the sentence according to your advice in L105-107. It now reads: “This study established a rat model of AD using STZ, and ASS was intraperitoneally injected for intervention.”
Point 4: In the Material and methods, it is needed to write information about how and from where Acanthopanax senticosus saponins (ASS) were received and from where STZ was received. Was a special extraction of Acanthopanax senticosus saponins done?
Author response: Labeling the brand and information of the drug in the Material and methods section is critical for other researchers in the future. According to your request, we have provided a detailed introduction to the methods and sources of obtaining Acanthopanax senticosus saponins (line142) and STZ (line136)
Point 5: In the statement "ASS can inhibit neuroinflammation and reduce the release of inflammatory factors, protecting nerve cells and alleviating AD," it is suggested to add a reference or use "may" instead of "can."
Author response: Thank you very much for your rigorous phraseology. We not only revised it in L294, but also checked and corrected linguistic expressions elsewhere in the manuscript.
Point 6: L369-372 please, add reference for this sentence “studies have reported that AS has various pharmacological activities, such as scavenging free radicals to increase superoxide dismutase activity in the 370 brain, inhibiting endothelin secretion and nitric oxide release, decreasing neuronal apoptosis during ischaemia, and increasing neuronal survival.”
Author response: We have added the relevant reference for this sentence in L393,
Reference 30

Reviewer 2 Report
Comments and Suggestions for Authors
The subject of the article is of great interest because it evaluates the neuroprotective effects of Acanthopanax senticosus saponins (ASS) in an experimental model of Alzheimer's disease (AD) induced in rats by streptozotocin (STZ). The study suggests that ASS could reduce neuroinflammation and Tau protein phosphorylation, improving cognitive abilities. The study is of great importance, considering the need for new therapeutic strategies for AD, and is structured with a multidisciplinary approach that includes behavioral tests, molecular and histological analyses. In my opinion, the use of the animal model (induced by STZ) is certainly valid but I believe that a limitation of this study is that it uses only one dose of ASS (50 mg/kg). In this way, it does not take into account the possible dose-dependent effects of ASS on the model. I suggest to evaluate whether different doses of ASS can have an improvement or a decrease in neuroinflammation and cognitive function. The authors also demonstrate that ASS treatment (50 mg/kg) significantly reduces inflammatory markers (IL-1β, TNF-α, NF-κB), DAPK1 expression and Tau protein phosphorylation, suggesting a possible mechanism of action via DAPK1-NF-κB and thus highlighting the therapeutic potential of ASS. Although the results are very interesting, it would be appropriate to include a control with a known drug (e.g. donepezil or others) in the study to better define the effects of ASS compared to currently used therapies. Furthermore, the STZ model has limitations in terms of reproducing the disease in patients. I would suggest integrating the STZ model with other approaches to confirm the results. Finally, to demonstrate the reduction of neuroinflammation through the hypothesized mechanism of action, I would recommend using selective inhibitors for NF-kB and DAPK1.
Author Response
Point 1: The study uses only one dose of ASS (50 mg/kg), not taking into account possible dose-dependent effects.
Author response: Thanks for your reminding, and we agree that evaluating different doses of ASS would be valuable. In fact, our previous pre-experimental studies found that ASS (50 mg/kg) is a suitable concentration according to the behavioral performance. Hence, in order to elucidate the mechanism of action of the drug as much as possible, we finished this study with the optimum concentration(50 mg/kg) of the drug. In the future, it is necessary for us to test the effect of different ASS concentration in treating STZ-induced AD.
Point 2: It would be appropriate to include a control with a known drug to better define the effects of ASS compared to currently used therapies.
Author response: As you mentioned, a control with known drug is beneficial to test the therapeutic effect of ASS. Unfortunately, due to certain limitations in our study design, we were unable to do so in this particular research. Although, the results of the behavioral and molecular experiments in this study are sufficient to show that ASS has significant therapeutic effects in STZ-induced AD. In future studies, we will follow your comments to make a good positive drug control, which will be more convincing and informative.
Point 3: The STZ model has limitations in terms of reproducing the disease in patients. The reviewer suggests integrating the STZ model with other approaches to confirm the results.
Author response: Thanks for your suggestions. We are aware of the limitations of the STZ model. In fact, we are currently utilizing 5xFAD mice to further clarify the efficacy of ASS in the treatment of AD after we complete the studies in this manuscript, which is of interest for future clinical treatment studies of ASS.
Point 4: To demonstrate the reduction of neuroinflammation through the hypothesized mechanism of action, the reviewer recommends using selective inhibitors for NF-kB and DAPK1.
Author response: Thanks for your advice. Using the STZ-induced AD model is only our preliminary study, and we are currently carrying out further experiments using 5xFAD mice, which involve utilizing selective inhibitors of NF-kB and DAPK1. In addition, we further utilized viruses to knock down the expression of NF-kB and DAPK1 genes. These experiments will provide more valid evidence for the findings of our current study.

Round 2
Reviewer 2 Report
Comments and Suggestions for Authors
The manuscript has certainly improved from the original version by adding more details and some more references. It can be published
Author Response
/